# Sex-Related Differences of Matrix Metalloproteinases (MMPs): New Perspectives for These Biomarkers in Cardiovascular and Neurological Diseases

**DOI:** 10.3390/jpm12081196

**Published:** 2022-07-22

**Authors:** Alessandro Trentini, Maria Cristina Manfrinato, Massimiliano Castellazzi, Tiziana Bellini

**Affiliations:** 1Department of Environmental and Prevention Sciences, University of Ferrara, Via Luigi Borsari 46, 44121 Ferrara, Italy; alessandro.trentini@unife.it; 2University Center for Studies on Gender Medicine, University of Ferrara, 44121 Ferrara, Italy; 3Department of Neuroscience and Rehabilitation, University of Ferrara, Via Luigi Borsari 46, 44121 Ferrara, Italy; mariacristina.manfrinato@unife.it (M.C.M.); tiziana.bellini@unife.it (T.B.); 4Interdepartmental Research Center for the Study of Multiple Sclerosis and Inflammatory and Degenerative Diseases of the Nervous System, University of Ferrara, 44121 Ferrara, Italy

**Keywords:** matrix metalloproteinases, cardiovascular disorders, central nervous system, sex-related differences, CNS, MMP

## Abstract

It is now established that sex differences occur in clinical manifestation, disease progression, and prognosis for both cardiovascular (CVDs) and central nervous system (CNS) disorders. As such, a great deal of effort is now being put into understanding these differences and turning them into “advantages”: (a) for the discovery of new sex-specific biomarkers and (b) through a review of old biomarkers from the perspective of the “newly” discovered sex/gender medicine. This is also true for matrix metalloproteinases (MMPs), enzymes involved in extracellular matrix (ECM) remodelling, which play a role in both CVDs and CNS disorders. However, most of the studies conducted up to now relegated sex to a mere confounding variable used for statistical model correction rather than a determining factor that can influence MMP levels and, in turn, disease prognosis. Consistently, this approach causes a loss of information that might help clinicians in identifying novel patterns and improve the applicability of MMPs in clinical practice by providing sex-specific threshold values. In this scenario, the current review aims to gather the available knowledge on sex-related differences in MMPs levels in CVDs and CNS conditions, hoping to shed light on their use as sex-specific biomarkers of disease prognosis or progression.

## 1. Introduction

Biologically based differences between the sexes impact several aspects of human life, starting from development to the construction of behaviour [1]. However, sex differences are also able to influence both health and disease, impacting biomedical research as well as individual and public health and healthcare delivery [2,3].

A remarkable aspect is that sex differences may influence susceptibility to a disease or its progression at both molecular and epidemiological levels [4]. This is evident by looking at the point global estimates for disability-adjusted life-year (DALY) disaggregated by sex, which represents both a measure of disease burden and a proxy for disease prevalence (Figure 1).

As displayed, virtually all conditions show a sex-driven disparity or a gender bias for injuries, but two conditions are of paramount importance when dealing with sex differences: cardiovascular diseases (CVDs) and neurological disorders. This is particularly true considering that CVDs are a leading cause of death, whereas neurological disorders strongly impact the quality of life of affected patients [5,6]. For these reasons, such conditions are constantly studied from a prevention and treatment point of view, and accumulating evidence suggests that both share an underlying sex difference [4]. As displayed in Figure 1, CVDs are characterized by a male-driven DALY, whereas neurological conditions (including stroke) present almost an equal proportion of both sexes. Although the exact causes for such differences are still largely unknown, the general knowledge about the matter in both disease groups is increasing. Despite this, a great deal of information is still lacking about the mechanisms of sex bias in most of the studies on biomarkers [7].

This is true also for matrix metalloproteinases (MMPs), enzymes responsible for extracellular matrix (ECM) turnover in both physiological states and diseases, that have been largely linked to both disorders [8,9]. For instance, high blood and cerebrospinal fluid levels of MMPs have been associated with an impaired blood–brain barrier (BBB) permeability, a fact that is directly correlated with increased infiltration of immune cells within the central nervous system (CNS) [10,11]. The same is true for CVDs, where high concentrations of MMPs have been related to atherosclerotic plaque instability and worse outcome [12]. Thus, considering the involvement of these important enzymes in such disorders, it is compelling to observe whether sex can impact MMPs expression in these diseases. To the best of our knowledge, no meta-analysis or updated review exists on this matter.

Therefore, in the attempt to fill in the gap, we aimed at gathering the available knowledge on potential sex-related differences in MMPs levels in CVDs and CNS conditions, hoping to shed light on their possible use as sex-specific biomarkers of disease prognosis or progression. To this end, after a brief introduction to the biochemistry of MMPs and their involvement in CVDs and CNS conditions, we will shift the focus to all the present information about sex differences.

## 2. Search Strategy and Data

We conducted a literature search in *MEDLINE* for all studies in the English language without age restriction, published any time to May 2022 with the keywords “MMPs” AND “neurodegenerative disorders” OR “neurodegenerative diseases” AND “sex difference” OR “gender difference”, in the title or abstract. The same criteria were followed for CVDs, with the keywords “MMPs” AND “cardiovascular disorders” OR “cardiovascular diseases” AND “sex difference” OR “gender difference”. To avoid missing possible articles on the matter, we repeated the literature search also separating by pathology. For neurodegenerative disorders, the keywords “Parkinson’s disease”, “Multiple Sclerosis”, “Alzheimer’s disease”, “Epilepsy”, “Ischemic stroke”, “Haemorrhagic stroke”, “Stroke”, “Migraine” were alternatively associated with AND “MMPs” AND/OR “sex difference”/“gender difference”. For CVDs, the keywords “Rheumatic Heart Disease”, “Hypertensive disease”, “Ischemic Heart disease”, “Coronary heart disease” OR “CAD”, “Cardiomyopathy”, “Myocarditis”, “Endocarditis”, “Myocardial infarction”, “Aneurism”, were alternatively associated with AND “MMPs” AND/OR “sex difference”/“gender difference”. We also searched for possible interesting articles on the matter by looking into reviews.

The data of global estimates of disability-adjusted life-year (DALY) disaggregated by sex and pathology were derived from the study Global Burden of Disease Study 2019 (GBD 2019) (https://vizhub.healthdata.org/gbd-results/, accessed on 12 July 2022). 

## 3. Matrix Metalloproteinases

As already said, MMPs are important zinc-dependent endoproteases involved in both physiological and pathological ECM remodelling [13]. For instance, they are involved in reproductive growth, embryonic development and morphogenesis during pregnancy, bone remodelling, tissue repair, and wound healing [14]. Being produced as zymogens, their net activity on tissues is a careful balance between activation processes and inhibition by specific inhibitors (tissue inhibitors of matrix metalloproteinases, TIMPs) and not specific ones (e.g., α2-macroglobulin) [15]. Thus, alterations in MMPs expression, activity/activation, and inhibition cause an accelerated ECM breakdown leading to a pathological condition, as happens in CVDs, musculoskeletal disorders, and various cancers [16,17,18,19].

MMPs have also been implicated in other processes such as systemic inflammation and CNS disorders, playing a pivotal role in the proteolytic degradation of the blood-brain barrier [10,11,20]. Collectively, circulating levels of MMPs have been proposed as potential markers of many cardiovascular and neurological diseases [8,9]. Before examining the potential influence of sex on the circulating levels of MMPs, we will briefly focus on the general structure of MMPs.

### 3.1. Structure and Function of MMPs

#### 3.1.1. General Structure and Regulation

This family of calcium- and zinc-dependent endopeptidases comprises more than 25 members divided into six classes based on their ability to degrade various components of the ECM: collagenases (MMP-1, MMP-8, MMP-13, MMP-18), gelatinases (MMP-2, MMP-9), stromelysins (MMP-3, MMP-10, MMP-11), matrilysins (MMP-7, MMP-26), membrane-type MMPs anchored to the cell membrane by a transmembrane (TM) domain or by a glycosylphosphatidylinositol (GPI)-anchored domain, and “other MMPs” [21,22,23]. Collagenases and gelatinases alter the molecules of the basal lamina, subsequently leading to cell death. In particular, collagenases degrade triple-helical fibrillar collagen in bone and ligaments, whereas gelatinases are involved in different cellular processes including angiogenesis and neurogenesis [14]. Stromelysins are small proteases that degrade segments of the ECM, and matrilysins process cell surface molecules and digest ECM components [14]. Finally, MT-MMPs have collagenolytic activity and may activate some proteases and components of the cell surface [24,25].

From a structural point of view (Figure 2), the MMP family shares a high homology, with a conserved zinc-binding motif (HEXXHXXGXXH) in the catalytic domain [26,27] and a common core structure consisting of a pro-peptide of about 80 amino acids, a catalytic domain of about 170 amino acids, a linker peptide called hinge region of variable length, and a hemopexin domain of about 200 amino acids. In addition, MMPs contain an amino-terminal signal sequence removed in the secretory pathway, targeting the enzyme to the endoplasmic reticulum (Figure 2, in grey) [28].

Four important divalent cations are embedded within the catalytic domain, possessing different roles: one zinc ion is coordinated by three histidines in the catalytic cleft and is essential in the polarization of water molecules necessary for the hydrolysis of the peptidic bond [28]. Within the same domain, another zinc ion and at least two calcium ions are essential to maintain the correct spatial structure for the interaction of substrates with the active site [28].

The hinge region (Figure 2, in dark blue) serves as a linking sequence to allow the free movement between the catalytic and hemopexin domains [28]. A carboxy-terminal (C-terminal) hemopexin-like (PEX) domain, with a four-bladed β-propeller structure connected to the hinge region, is usually involved in substrate recognition and inhibitor binding [29]. Additionally, different classes of MMPs have peculiar structural features that distinguish them from the prototypical MMP structure: for instance, gelatinases (MMP -2 and MMP-9) have a fibronectin-like domain inserted within the catalytic domain that facilitates both gelatin and collagen binding [30].

MMPs are maintained into an inactive form by a pro-peptide domain, which should be removed to convert the enzyme into the active form, an event that usually happens in the extracellular space through other proteolytic enzymes such as serine proteases, plasmin, or other MMPs [14]. As an exception, some MMPs have a furin recognition site before the catalytic domain, allowing the intracellular activation of the enzyme by furin [10,21].

Other pathophysiological activation pathways include the modification of the thiol group of the cysteine in the pro-domain and are responsible for maintaining the enzyme inactive by reactive oxygen species (ROS) and reactive nitrogen species (RNS). Such modification leads to the oxidation of the side chain of the amino acid with S-nitrosylation or S-glutathionylation being the predominant changes [31,32]. 

The expression and activity of MMPs are regulated at many levels. At the transcriptional level, the synthesis of new MMPs is regulated by various cytokines, growth factors, and ROS [33].

Once synthesized, the activity of MMPs depends on the activation of their latent form as well as on endogenous tissue inhibitors, namely α2-macroglobulins and TIMPs [22], which bind to the enzymes suppressing the activity. When the activity is not well-balanced by a fine-tuned inhibition, a transition from physiological to pathological condition occurs [14].

Currently, four TIMPs have been described (from TIMP-1 to TIMP-4), with TIMP-1 and TIMP-2 being the most studied. Interestingly, TIMP-2 can participate in both inhibition and activation of enzymes by forming 1:1 stoichiometric non-covalent complexes [34].

A particular mention must be made for MMP-9, which belongs to the gelatinase subfamily. While the inactive form of MMP-9 (proMMP-9) is usually secreted in a complex with TIMP-1 [35], it also exists in a TIMP-1 free form released by neutrophils [36]. In addition, there are at least two active forms of MMP-9 in body fluids: an N-truncated active enzyme that can be regulated by TIMP-1 and a “fully activated” enzyme lacking both the N- and C-terminal hemopexin domains, which cannot be inhibited by TIMP-1 at physiological concentrations [37,38].

#### 3.1.2. Role of MMPs in CVDs

As said before, MMPs can be largely modulated by cytokines and growth factors, molecules that are produced during inflammatory conditions. As expected, inflammation is a cornerstone in most of the CVDs [39] and acts by deranging the delicate axis between MMP activation and inhibition. For instance, MMP activation can alter the architecture of atherosclerotic plaque helping its disruption [40]. However, their role is not just relegated to the “simple” plaque instability, but they also participate in a plethora of cardiovascular conditions including aneurism, myocardial infarction, atherosclerosis, hypertension, and cardiomyopathies (Table 1).

In aneurism, MMP-1, MMP-2, MMP-3, MMP-8, and MMP-9 are overexpressed within the aortic wall, participating in the weakening of ECM structure through the degradation of several collagens, elastin, and fibronectin [49,50].

A role in myocardial infarction (MI) has been found as well although is still far to be clear and somewhat complicated. MMPs seem to be involved in both healing processes after MI injury [57,58] as well as in the adverse remodelling that occurs after MI [52,59,60,61]. Interestingly, if the cytokine production following MI is controlled, an appropriate wound healing response is established, where MMPs, especially MMP-9 and MMP-2, actively participate in the process by degrading the ECM, favouring the infiltration of inflammatory cells and processing cytokines and chemokines [51]. However, if the production of MMPs is actively sustained well beyond the first 72 h (the period after the initial wound healing is complete), their action could be harmful [62].

One pathogenic mechanism by which MMPs can contribute to atherosclerosis is through the enhancement of vascular smooth muscle cell migration and the formation of neointima after a vascular injury [63,64]. In addition, as said before, active MMPs produced within the atherosclerotic lesion may contribute to plaque instability: in particular, MMP-2, MMP-3, MMP-8, MMP-9, and MMP-12 have been connected to plaque development and a worse outcome [52,53,54,55,56]. 

A connection has also been found between MMPs and hypertension [41]. In particular, MMP-2 was increased in the thoracic aortas and heart of a model of hypertensive rat, where it may participate in hypertension-induced vascular remodelling [42,65,66]. This may be partially mediated by the ability of some MMPs to cleave receptors and factors responsible to maintain the vascular tone. For instance, MMP-7 and MMP-9 injection in spontaneously hypertensive rats (SHR) causes vasoconstriction, thereby increasing the blood pressure [43]. This may be due to a not yet well-characterized increase in MMP-dependent β2-adrenergic shedding. In addition, in vitro studies have found that MMP-2 can cleave factors that can increase vasoconstriction activity [67] or limit their vasodilating effect [44,45].

Finally, a growing body of evidence points toward MMP-2 as a major player in cardiomyopathies. For instance, the overexpression of active MMP-2 alone in the heart of transgenic mice models of ischemia/reperfusion can increase the infarct area and decrease the contractile function of the hearth [46]. On the contrary, the deletion of MMP-2 seems to be protective toward the heart remodelling and functionality following injury [68]. This might be at least partially related to the ability of MMP-2 to process several cardiac proteins including troponin I and myosin light chain [47,48].

#### 3.1.3. Role of MMPs in Neurological Conditions

Within the CNS, MMPs participate both in (i) physiological processes such as neurogenesis, axonal guidance, synaptic plasticity, learning, and memory [69] and (ii) in pathological conditions such as neuroinflammation, neurodegeneration, and cerebrovascular related disorders [70].

Matrix metalloproteinases can be produced by several CNS-related cells, including endothelial cells, microglia, oligodendrocytes, neurons, and astrocytes [71]. Under normal conditions, MMPs have been observed to be (i) generally absent or (ii) at undetectable or (iii) otherwise modest levels in the mature brain [20]. A dysregulation of MMP activity and/or the presence of upregulatory stimuli could alter the physiological balance, inducing a pro-inflammatory state [72,73].

The MMPs most involved in brain processes are MMP-2, MMP-3, MMP-9, MMP-10, and MMP-14 [74,75] (Table 2).

While MMPs are considered generally absent, or in any case scarcely detectable in the CNS, MMP-2 seems to be an exception because it is physiologically expressed also in the healthy brain and cerebrospinal fluid (CSF) [74].

In pathological conditions, MMPs may increase blood–brain-barrier (BBB) permeability by acting on the basal lamina and tight junctions in endothelial cells, resulting in the final common pathway of acute neuroinflammatory damage [74,87].

In Alzheimer’s disease (AD), MMP-2 might be assumed to have a protective role [76], and in MS, active MMP-2 was associated with the remission phase of the disease, suggesting a role of this enzyme in the termination of MS neuroinflammation [77].

If, on the one hand, MMP-3 was physiologically associated with synaptic plasticity, learning, and neuronal development, in the other, uncontrolled MMP-3 activity was associated with Alzheimer’s and Parkinson’s diseases (AD and PD) [78,88].

MMP-9 is probably the most studied MMP in CNS disorders, especially in association with neuroinflammation. MMP-9 was associated with demyelination in MS [80], neuronal cell death in AD [81], neuroinflammation in PD [20], dolichoectasia [82,83], and neuroinflammation and cell death in amyotrophic lateral sclerosis (ALS) [84,85]. In hypoxic-ischemic lesions, MMP-2, MMP-3, and MMP-9 together increase the permeability of the BBB with a consequent greater risk of haemorrhagic transformation [74].

Finally, MMP-10 inhibition cleaves huntingtin and reduces cell death [86].

## 4. Sex Differences in MMPs and Pathologies

### 4.1. Sex Differences in MMPs and Cardiovascular Diseases

Cardiovascular diseases have long been seen as conditions that primarily affect males [4]. This conclusion can be drawn also by looking at Figure 1, where males account for almost 58% of the CDV group of DALY. However, the age-corrected risk of CVDs is similar between males and females [89] since it becomes equal between sexes after menopause or even increases for females. Nonetheless, by separating CVDs into several categories, we can see that the situation is more complex. Indeed, ischemic heart disease; the spectrum of disorders including cardiomyopathy, myocarditis, and endocarditis; and aneurism are far more frequent in males than females (Figure 3). On the other hand, females seem primarily affected by rheumatic and hypertensive heart disease (Figure 3).

Despite this evidence and the impact that sex hormones have on the expression of MMPs [79], the reports about a cross-interaction between MMPs, CVDs, and sex are still in their infancy. According to what is present in the literature, the most studied MMPs showing a sex difference in CVDs are MMP-2, MMP-3, MMP-9, MMP-8, MMP-13, and MMP-14 although the information on the last two members is scarce (see below for detailed information).

The results about sex disparity of MMP-2 in CVDs are conflicting and at variance with the considered condition. For instance, male patients affected by aneurysms have shown increased levels of MMP-2 in the aorta [90], a fact confirmed also in vivo in the mouse model [91] and in vitro on rat smooth muscle cells [92]. However, the opposite was found in females when considering patients with bicuspid valve thoracic aortic aneurysm (BV-AAA) [93,94]. This apparent contradiction could be explained by the fact that females affected by BV-AAA also show an extensive remodelling of the aorta [95], which is characterized by a decrease in deposition of ECM components and by increased degradation operated essentially by MMPs [96]. Finally, the levels of the MMP-2 were found elevated in the hearts of male rats during acute myocardial infarction (MI) [97], with higher serum levels of the enzyme found to be associated with lower myocardial fibrosis only in females [98].

MMP-3 is one of the enzymes that demonstrated an undeniable increase in males with respect to females in most of the studies dealing with MI [99,100,101]. In addition, serum MMP-3 was positively correlated with inflammatory markers only in females [101]. Collectively, these results may suggest, at least partly, a different interaction between sex and inflammation in the etiopathogenesis of MI [101].

The reports on MMP-8 (a.k.a. called neutrophil collagenase), which has specificity towards type I, II, and III collagens, are few, and all found decreased levels of the enzyme or no change in females suffering from thoracic aortic aneurysm [94] or in atherosclerotic plaques [102,103], suggesting a protective effect of sex hormones in females, at least in aneurysm [104].

MMP-9 is far, together with MMP-2, the most studied metalloprotease in all aspects of CVDs, especially considering its connection with inflammation [105]. Most of the studies present in the literature have found an increase in the ascending aorta of male patients affected by aneurysms [90,93,106], with only one report finding the opposite result in females [91]. The former results are in large agreement with the clinical observation of a higher prevalence of aneurysms in males, making it a striking example of male disadvantage in CVDs [104].

In MI, MMP-9 has been found to increase in the heart of male rats and mice when compared to females [97,107] and is associated with increased rupture of the left ventricle [108]. Although the fibrotic event seems not associated with different levels of the enzyme between the sexes, only in females are high MMP-9 serum concentrations correlated with markers of myocardial fibrosis [98].

Human coronary disease patients show contrasting results, with reports suggesting an increase in serum of males [109,110], whereas others found the opposite in females [111,112]. Collectively, it is undoubted that MMP-9 plays a key role in the aetiology of several CVDs, and it may point toward a different use of this protein as a sex-specific biomarker.

The preliminary results about MMP-13 and MMP-14 found in humans and confirmed in rats suggest an increased activity and production of both proteins in males affected by aneurism [94,113]. This is consistent with an increased ECM degradation and weakened aortic wall that makes males more prone to rupture. Again, this confirms the male disadvantage, present in certain cardiac conditions, over females.

The studies on sex-driven MMP expression in myocarditis are preliminary, and the only work we found in mice observed an increase in MMP-8 expression in the heart during myocarditis [114]. This may have an implication for myocardial remodelling and fibrosis during this condition [114]. Regarding rheumatic heart disease, despite several works found an increased expression of MMP-1 and MMP-9 in the mitral valve [115] and/or serum [116,117,118] of patients compared to controls, correlated with the underlying inflammation, there is currently no single study that explored a possible influence of sex on MMP expression. Sex-related differences in MMPs in patients affected by CVDs are summarized in Table 3.

### 4.2. Sex Differences in MMPs and CNS Disorders

Sex differences in the CNS are quite evident and touch several aspects of the brain. For instance, sex hormones affect brain anatomy in a region-specific manner, by influencing neuronal growth and development [123]. The brain is also affected at the biochemical/neurochemical level. As an example, females synthesize far less serotonin than their male counterparts, making them more susceptible to depression [124]. In addition, metabolic differences between males and females have been found in several regions of the brain [125]. Finally, various psychological and cognitive processes can be influenced as well [126].

The fundamental sex differences found in the anatomy, biochemistry, and genetics of a healthy brain translate also into a sex-driven disparity in susceptibility, progression, symptom severity, and pathology of disorders affecting the CNS [127]. This is clear also by looking at Figure 4. Indeed, most neurological conditions are characterized by a female-shifted disease burden, underlying an increased prevalence of neurological diseases in this sex. The only exceptions appear to be PD, epilepsy, haemorrhagic stroke, and ALS, where a male prevalence is documented [128,129,130,131]. Of note, the apparent counterintuitive lack of difference in neurological conditions shown in Figure 1 is mainly due to the inclusion of stroke-related diseases in this group. In fact, by leaving stroke within the CVDs, the sex bias in neurological conditions appears to be more evident (DALYs*100,000 population (95% Uncertainty Level), males: 413 (251–646); females: 565 (303–953)). For this reason, it is of paramount importance that the effect of sex is evaluated also in each condition separately to avoid possible misinterpretations driven by grouping variables.

From the data reported in the literature, it seems clear that one of the major implications of MMPs in CNS disorders occurs at the level of regulation of the BBB [10]. It is interesting to highlight how previous studies have demonstrated the ability of the hormone 17β-oestradiol to reduce the degradation of tight junction proteins by suppressing the upregulation of MMP expression [74], thus making oestrogens play a protective role towards the degradation of BBB [132]. From a clinical point of view, a growing number of articles are highlighting the existence of a sexual dimorphism in the CSF protein content resulting in higher levels in men than in women regardless of age [133,134]. The cause of this dimorphism seems to be attributable to a greater permeability to plasma proteins, especially albumin and IgG, of the blood–CSF barrier (BCSFB), as has been shown in patients with MS and patients with other inflammatory and non-inflammatory neurological disorders [135], in psychiatric patients [136], in subjects to whom lumbar puncture was performed for diagnostic purposes, and in healthy subjects [137].

A difference between sexes in BBB permeability was also confirmed using dynamic contrast-enhanced (DCE)-magnetic resonance imaging (MRI) [138]. In that study, the authors demonstrated a better BBB integrity in cingulate and occipital cortices in females than in male non-demented elderly subjects. They also found that this sex-related difference in BBB integrity was attenuated by aging or when cognitive decline occurred, but the difference remained in the occipital cortex independently of these two factors [138].

Although currently few studies in the literature have analysed MMPs as a function of sex in patients affected by neurological disorders, some interesting data have already emerged.

Reduced serum levels of MMP-1 were found in PD patients in comparison to controls, and this difference was more evident in females [139]. On the contrary, CSF levels of MMP-1 were higher in females than in males in people with MS and neurological controls [140].

While high levels of MMP-3 appear to be more associated with male sex in AD patients in two studies [141,142], an increase in MMP-3 was associated with greater cognitive impairment in females [141]. Moreover, serum levels of MMP-3 were higher in male subjects with epilepsy with respect to females [143].

High levels of serum MMP-9 were more likely found in male subjects during the acute phase of ischemic stroke and were associated with an increased risk of major disability and death [144].

Sex-specific patterns of MMP expression were described in the days following intracerebral haemorrhage; in particular, MMP-3 and MMP-10 acted as predictors of long-term functional outcomes in male and female patients, respectively [145].

Finally, higher CSF MMP-10 levels were found in female than in male MS patients [140].

Sex-related differences in MMPs in patients affected by CNS disorders are summarized in Table 4.

## 5. Conclusions

The importance of studying sex differences lies in the fact that they are an important factor contributing to individual differences. This stemmed from several epidemiological and observational studies, including those on cardiovascular and neurological disorders. For instance, males have a higher incidence of stroke across much of their lifespan, but after 80 years of age, the situation is reversed, disadvantaging females. Keeping stroke as an example, symptoms might be sex-specific, and the type of stroke might occur with a sex bias. Therefore, in this current era of precision medicine, it is of paramount importance to consider sex differences not only related to aetiology but also at the biomarkers’ level.

In fact, from what we found (see Figure 5 for a summary), there is a clear disparity between males and females in MMP levels in several disorders. However, it is important to acknowledge that in a large part of the studies, sex has typically been considered a confounding factor used to correct statistical models to solely relate the biomarkers to the disease. As such, sex-specific differences have not been examined, eliminating part of the biological information that might help in identifying novel disease patterns or new sex-specific disease biomarkers. Therefore, it is compelling to evaluate sex differences in each study, especially considering that just because a difference is not apparent, it does not mean that is not visible. More importantly, sex differences should be studied across the lifespan of individuals.

Thus, the consideration of sex differences in MMPs and more in general in biofluid-based biomarkers is important and timely for their future application to clinical practice.

Therefore, a systematic disaggregation and analysis of data by sex may shed light on the possible use of MMPs as sex-specific biomarkers of disease prognosis and progression.

## Figures and Tables

**Figure 1 jpm-12-01196-f001:**
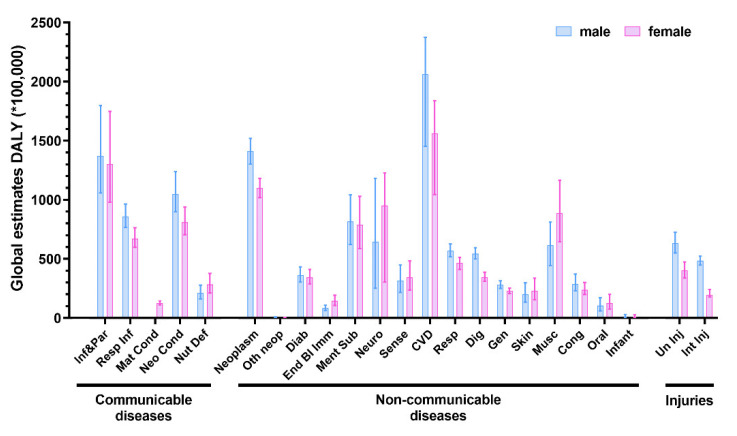
Global estimates of disability-adjusted life-year (DALY) are disaggregated by sex for communicable diseases, non-communicable diseases, and injuries. Source: Global Burden of Disease Study 2019 (GBD 2019) (https://vizhub.healthdata.org/gbd-results/, accessed on 12 July 2022). Inf&Par, infectious and parasitic diseases; Resp inf, respiratory infectious; Mat Cond, maternal conditions; Neo Cond, neonatal conditions; Nut Def, nutritional deficiencies; Oth neop, other neoplasms; Diab, diabetes mellitus; End Bl Imm, endocrine, blood, immune disorders; Ment Sub, mental and substance use disorders; Neuro, neurological conditions; Sense, sense organ diseases; CVD, cardiovascular diseases; Resp, respiratory diseases; Dig, digestive disease; Gen, genitourinary diseases; Skin, skin diseases; Musc, musculoskeletal diseases; Cong, congenital diseases; Oral, oral conditions; Infant, sudden infant death syndrome; Un Inj, unintentional injuries; Int Inj, intentional injuries. Error bars represent the 95% uncertainty level (95% UL).

**Figure 2 jpm-12-01196-f002:**
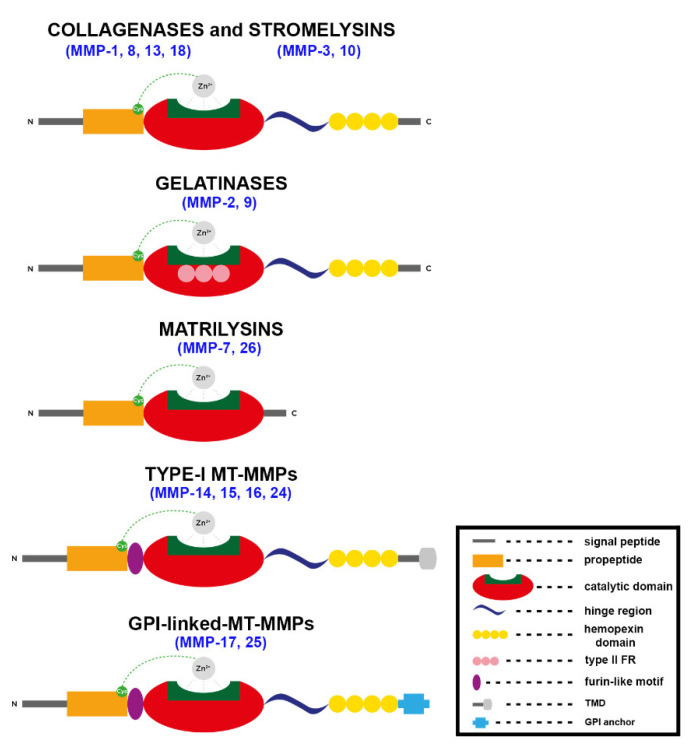
Schematic structure of the major classes of MMPs. FR, fibronectin repeats; TMD, transmembrane domain; GPI, glycosylphosphatidylinositol. The structure of MMP-11 is similar to that of stromelysins except for a furin-like domain between the propeptide and the catalytic domain.

**Figure 3 jpm-12-01196-f003:**
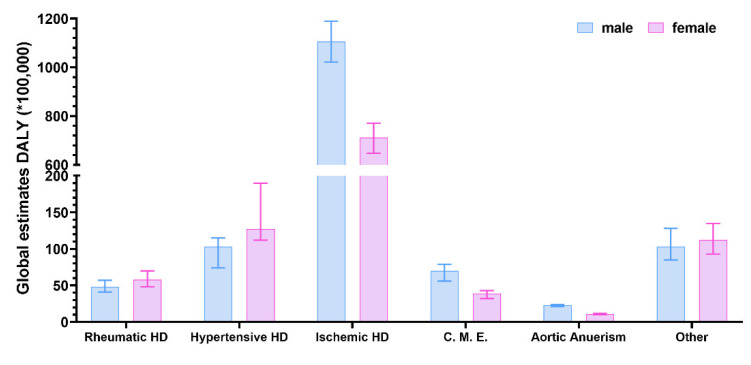
Global estimates of disability-adjusted life-year (DALY) disaggregated by sex for cardiovascular diseases (CVDs). HD, heart disease; C.M.E., cardiomyopathy, myocarditis, endocarditis. Source: Global Burden of Disease Study 2019 (GBD 2019) (https://vizhub.healthdata.org/gbd-results/, accessed on 12 July 2022). Error bars represent the 95% uncertainty level (95% UL).

**Figure 4 jpm-12-01196-f004:**
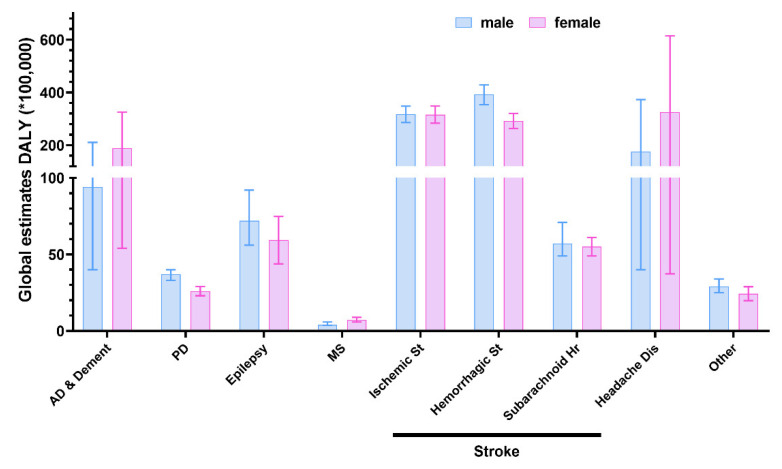
Global estimates of disability-adjusted life-year (DALY) disaggregated by sex for neurological conditions. AD & Dement, Alzheimer’s disease AND other dementias; PD, Parkinson’s disease; MS, multiple sclerosis; St, stroke; N-M Head, non-migraine headache; Other, other neurological conditions. Source: (https://vizhub.healthdata.org/gbd-results/, accessed on 12 July 2022).

**Figure 5 jpm-12-01196-f005:**
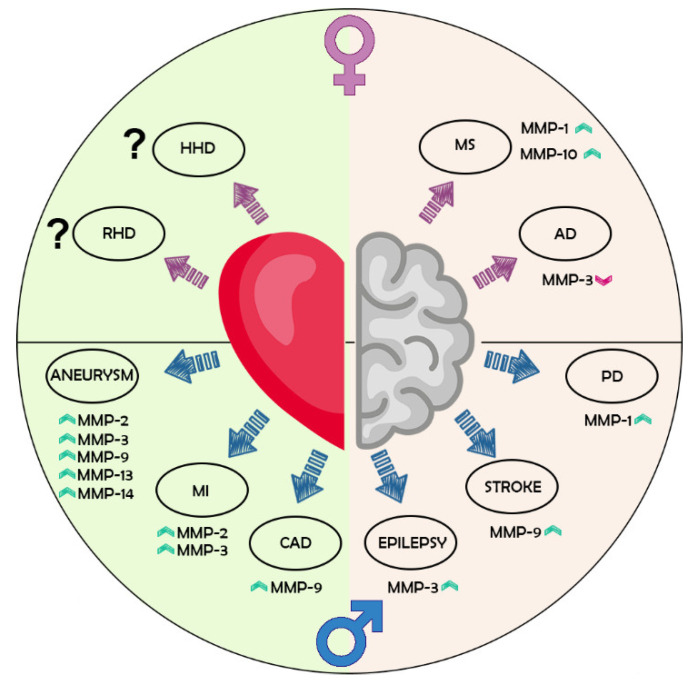
Summary of the main changes in MMPs separated by sex and pathology. As pictured, on the top are represented the diseases more frequent in females, while on the bottom are listed those with an increased prevalence in males. To make the summary clearer, neurological conditions (right side, light salmon colour sector) are separated by CVDs (left side, light green sector). For instance, MS and AD, which are neurological conditions more frequent in females than males, are localized in the top right of the picture. While females with MS demonstrated to have increased levels of both MMP-1 and MMP-10, females with AD have lower levels of MMP-3. On the contrary, males with PD, stroke, and epilepsy (right bottom side), have higher levels of MMP-1, MMP-9, and MMP-3, respectively. The situation is different for CVDs, where there is a more male-driven sex disparity, with aneurysm, MI, and CAD more frequent in this sex (left bottom). As such, several MMPs have been found to increase in males affected by these conditions. Finally, although HHD and RHD seem to be more frequent in females (top left side), results about MMP levels are still lacking (question mark symbol). HHD, hypertensive heart disease; RHD, rheumatic heart disease; MS, multiple sclerosis; AD, Alzheimer’s disease; PD, Parkinson’s disease; CAD, coronary heart disease; MI, myocardial infarction.

**Table 1 jpm-12-01196-t001:** Matrix metalloproteinases (MMPs) that are mainly involved in cardiovascular diseases (CVDs).

MMPs	CVDs	Role	References
MMP-2	Hypertension	Causes vasoconstriction	[41,42,43]
		Limits vasodilation	[44,45]
	Cardiomyopathies	Increases infarct areas	[46]
		Process several cardiac proteins	[47,48]
MMP-1, -2, -3, -8, and -9	Aneurism	Weakens the ECM structure	[49,50]
MMP-2 and -9	Myocardial infarction	Degrade ECM/process cytokines and chemokines	[51]
MMP-2, -3, -8, -9, and 12	atherosclerosis	connected to plaque development	[52,53,54,55,56]
MMP-7 and -9	hypertension	cause vasoconstriction/increase blood pressure	[43]

ECM, extracellular matrix.

**Table 2 jpm-12-01196-t002:** Matrix metalloproteinases (MMPs) that are mainly involved in central nervous system (CNS) pathologies.

MMPs	CNS Conditions	Role	References
MMP-2	AD	Protective	[76]
	MS	Associated to remission	[77]
MMP-3	AD and PD	Detrimental	[78,79]
MMP-9	MS	Associated to demyelination	[80]
	AD	Associated to neuronal cell death	[81]
	PD and dolichoectasia	Associated to neuroinflammation	[20,82,83]
	ALS	Associated to neuroinflammation	[84,85]
MMP-10	HD	Detrimental	[86]
MMP-2, -3, -9	HI lesions	Increase BBB permeability	[74]

AD, Alzheimer’s disease; ALS, amyotrophic lateral sclerosis; BBB, blood–brain-barrier; HD, Huntington’s disease; HI, hypoxic ischemic; ICH, intracerebral haemorrhage; MS, multiple sclerosis; PD, Parkinson’s disease.

**Table 3 jpm-12-01196-t003:** Sex-related differences in matrix metalloproteinases (MMP) in different cardiovascular diseases (CVDs).

MMPs	CVD	Organism	Male	Female	References
MMP-2	Aneurism	Human/Mouse	↑ in aorta		[90,91]
	Thoracic aortic aneurysm	Human		↑ in aorta	[93,94]
	Acute MI	Mouse	↑ in affected heart		[97]
	Heart failure	Human		↓ in serum	[119]
	Hypertension	Human	No change	No change	[120]
	Myocardial fibrosis	Human		↑ serum concentration associated with LOWER fibrosis	[98]
MMP-3	MI	Human		↓ in serum; positively correlated with inflammatory markers	[99,100,101]
MMP-8	Thoracic aortic aneurysm	Human		↓ in aorta	[94]
	Carotid atherosclerosis	Human	No difference	No difference; ↓ in plaques. The difference disappeared in multivariate correction	[102,103]
MMP-9	Aneurism/Ascending thoracic aortic aneurysms	Human/Mouse	↑ in the ascending aorta	↑ in aorta	[90,91,93,106]
	MI	Mouse/Rat	↑ in heart	↓ in hearth	[107,97]
	post MI	Human/Mouse	No change/Positively correlated with inflammatory markers; ↑ in Heart	No change	[121,101]
	Cardiac rupture following MI	Mouse	↑ in left ventricle		[108]
	Myocardial fibrosis	Human		↑ serum concentration associated with HIGHER fibrosis	[98]
	CAD/CHD	Human	↑ in serum	↓ in serum	[109,110,111,112]
	Chest Pain	Human		↑ in serum (patients with non-calcified and mixed plaques)	[122]
	Hypertension	Human	No change	No change	[120]
MMP-13	Thoracic aortic aneurysm	Human/Rat		↓ in aorta	[94,113]
MMP-14	Thoracic aortic aneurysm	Human	↑ in aorta		[94]

CAD, coronary artery disease; CHD, coronary heart disease; MI, myocardial infarction; ↑ denotes an increase, ↓ a decrease.

**Table 4 jpm-12-01196-t004:** Sex-related differences in matrix metalloproteinase (MMP) levels in different conditions of the central nervous system.

MMPs	CNS Conditions	Male	Female	References
MMP-1	PD		↓ in serum	[139]
	MS and neurological controls		↑ CSF levels	[140]
MMP-3	AD, MCI, and cognitively normal individuals	↑ frontal cortex protein levels		[142]
		↑ plasma levels	↑ levels associated to greater cognitive decline	[141]
	Epilepsy	↑ serum levels		[143]
	ICH	associated to long-term functional outcomes		[145]
MMP-9	Ischemic stroke	↑ serum levels in acute phase		[144]
MMP-10	ICH		Associated to long-term functional outcomes	[145]
	MS		↑ CSF levels	[140]

CNS, central nervous system; AD, Alzheimer’s disease; CSF, cerebrospinal fluid; ICH, intracerebral haemorrhage; MCI, mild cognitive impairment; MS, multiple sclerosis; PD, Parkinson’s disease; ↑ denotes an increase, ↓ a decrease.

## Data Availability

The data used for this study are publicly available to download from https://vizhub.healthdata.org/gbd-results/.

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
