# Peer review of "Sex-Related Differences of Matrix Metalloproteinases (MMPs): New Perspectives for These Biomarkers in Cardiovascular and Neurological Diseases"

_jpm, 2022, doi:10.3390/jpm12081196_

Round 1

Reviewer 1 Report

“ Sex-related differences of Matrix Metalloproteinases (MMPs): new perspectives for these biomarkers in cardiovascular and neurological disease”

The review is interesting and clear.

My comments are a follows:

A brief paragraph regarding the methods used for this review should be added (even if it is a narrative review). Thus, database used, criteria and timing of selection etc should be specified.

Lines 33-41, Lines 78-80: References should be provided.

There are several sentences throughout the manuscript without references. The authors should check add references where appropriate.

Figure 1: Is it possible to add statistical analysis in order to understand where the difference (male vs female) is significant?

Figure 2: It should be improved. For example, the classes of MMPs could be indicated.

Section 2.1.1: the role of calcium and zinc could be better explained.

Section 2.1.2: I suggest to add a table to summarize the role of the different MMPs in CVDs.

Section 2.1.3: A table summarizing the roles of MMPs in neurological conditions would be helpful.

Figures 3 and 4: Again, statistical analysis would help in understanding the significant differences.

Lines 321-323: the authors reported that the studies on MMPs expression in myocarditis and rhematic heart disease are preliminary and there is very little to no information available in the current literature at this regard.” Could the authors add the preliminary information/what is known about these points? or at least add the relative references.

Line 355: Which MMP?

Lines 410-413: references should be added.

Caption of figure 5 should be improved.

Abbreviations should be defined at first mention and used consistently throughout the manuscript (for example, line 158: “reactive oxygen species” should be “ROS”)

Author Response

Reviewer 1

The review is interesting and clear.

My comments are as follows:

A brief paragraph regarding the methods used for this review should be added (even if it is a narrative review). Thus, database used, criteria and timing of selection etc should be specified.

R: We thank the Reviewer for this suggestion. We added a paragraph explaining the search criteria in the new version of the manuscript (see lines 94-113), and reported below for convenience.

2.0. Search strategy and statistics

We conducted a literature search in MEDLINE for all studies in the English language without age restriction, published any time to May 2022 with the keywords “MMPs” AND “neurodegenerative disorders” OR “neurodegenerative diseases” AND “sex difference” OR “gender difference”, in the title or abstract. The same criteria were followed for CVDs, with the keywords “MMPs” AND “cardiovascular disorders” OR “cardiovascular diseases” AND “sex difference” OR “gender difference”. To avoid missing possible articles on the matter, we repeated the literature search also separating by pathology. For neurodegenerative disorders, the keywords “Parkinson’s disease”, “Multiple Sclerosis”, “Alzheimer’s disease”, “Epilepsy”, “Ischemic stroke”, “Hemorrhagic stroke”, “Stroke”, “Migraine”, were alternatively associated with AND “MMPs” AND/OR “sex difference”/“gender difference”. For CVDs, the keywords “Rheumatic Heart Disease”, “Hypertensive disease”, “Ischemic Heart disease”, “Coronary heart disease” OR “CAD”, “Cardiomyopathy”, “Myocarditis”, “Endocarditis”, “Myocardial infarction”, “Aneurism”, were alternatively associated with AND “MMPs” AND/OR “sex difference”/“gender difference”. We also searched for possible interesting articles on the matter by looking into reviews.

The data of global estimates of disability-adjusted life year (DALY) disaggregated by sex and pathology were derived from the study Global Burden of Disease Study 2019 (GBD 2019) (https://vizhub.healthdata.org/gbd-results/). 

Lines 33-41, Lines 78-80: References should be provided.

  1. As suggested by the Reviewer, we added the following references (here are just the numbers of the new references): [1], [2], [3], [7], [8], [9]

There are several sentences throughout the manuscript without references. The authors should check/add references where appropriate.

  1. We thank the Reviewer for pointing this out. We checked the text and added references to sentences where appropriate.

Here is the list of added references (just the new numbers): 14, 35, 36, 86, 87, 107, 122, 123, 124, 125, 126, 136.

Figure 1: Is it possible to add statistical analysis in order to understand where the difference (male vs female) is significant?

  1. We thank the Reviewer for his/her suggestion. However, given the nature of the measurement, we can’t make an inference between the two groups, male and female. DALY, which is the sum between YLL (years of life lost) and YLD (years lived with disability), can be considered as a punctual measurement (not a mean) where each available country worldwide contributes to the final measure, with opportune weights. The presented estimate has been obtained through modeling and bootstrapping with 1000 draws. In addition to the values, we were able to obtain the 95% uncertainty level (95% UI), which was created from the 0.025 and 0.975 quantiles of the draws. Detailed information about the whole procedure can be found in “GBD 2019 Diseases and Injuries Collaborators. Global burden of 369 diseases and injuries in 204 countries and territories, 1990-2019: a systematic analysis for the Global Burden of Disease Study 2019. Lancet. 2020 Oct 17;396(10258):1204-1222. DOI: 10.1016/S0140-6736(20)30925-9. Erratum in: Lancet. 2020 Nov 14;396(10262):1562. PMID: 33069326; PMCID: PMC7567026”.

According to what we said, we changed the figures by substituting the relative percentage with the absolute values of DALYs and adding the 95% UL.

Figure 2: It should be improved. For example, the classes of MMPs could be indicated.

  1. As the reviewer suggested, we changed Figure 2 and added the classes of MMPs.

Section 2.1.1: the role of calcium and zinc could be better explained.

  1. We thank the Reviewer for this suggestion. We added a sentence to better explain the role of calcium and zinc as follows (see lines 166-171): “Four important divalent cations are embedded within the catalytic domain, possessing different roles: one zinc ion is coordinated by three histidines in the catalytic cleft, and is essential in the polarization of water molecules necessary for the hydrolysis of the peptidic bond [28]. Within the same domain, another zinc ion and two calcium ions are essential to maintain the correct spatial structure for the interaction of substrates with the active site [28].”

Section 2.1.2: I suggest to add a table to summarize the role of the different MMPs in CVDs.

  1. We thank the reviewer for this suggestion. Accordingly, a table summarizing the role of MMPs in CVDs was added. See the new Table 1 (line 218 for its citation).

Section 2.1.3: A table summarizing the roles of MMPs in neurological conditions would be helpful.

  1. We thank the reviewer for his/her suggestion. Accordingly, a table summarizing the role of MMPs in neurological conditions was added. See Table 2 (line 269 for its citation).

Figures 3 and 4: Again, statistical analysis would help in understanding the significant differences.

  1. As already replied to point 4 about figure 1, currently a statistical comparison between the two sexes is not possible with the available data and given the nature of DALY estimation.

Lines 321-323: the authors reported that the studies on MMPs expression in myocarditis and rheumatic heart disease are preliminary and there is very little to no information available in the current literature in this regard.” Could the authors add the preliminary information/what is known about these points? or at least add the relative references.

  1. We thank the Reviewer for this helpful comment. As we stated in the text, the results about sex influence on MMP in myocarditis are very preliminary, and we were able to find only one work. In this work, it is claimed that:

1) myocarditis is more frequent in men than women;

2) testosterone replacement in gonadectomized mice caused an increase in MMP-8 expression in the heart, during myocarditis (ref. 122). Of course, this may have implications for myocardial remodeling.

Regarding rheumatic heart disease, there are works only showing an association of several MMPs, such as MMP-1 in the mitral valve [ref 123] and the serum of patients compared to controls [ref 124], MMP-9 in serum of patients [ref 126, 126] with the disease, but none dealing with a possible sex influence on MMP levels.

Since there is only one work and it has not been replicated, we decided to exclude this information from Table 2 and Figure 5.

However, we included ot in the text as follows (see lines 364-368): “Regarding rheumatic heart disease, despite several works found an increased expression of MMP-1 and MMP-9 in the mitral valve [123] and/or serum [124–126] of patients compared to controls, correlated with the underlying inflammation, there is currently not a single study that explored a possible influence of sex on MMP expression.”

Line 355: Which MMP?

  1. We thank the reviewer for his/her comment. Accordingly, we added the appropriate reference.

Lines 410-413: references should be added.

  1. Done, reference [136].

Caption of figure 5 should be improved.

  1. We thank the Reviewer for this suggestion. Now the caption of Figure 5 has been made clearer.

“Summary of the main changes in MMPs separated by sex and pathology. As pictured, on the top are represented the diseases more frequent in females, while on the bottom are listed those with an increased prevalence in males. To make the summary clearer, neurological conditions (right side, light salmon color sector) have been separated by CVDs (left side, light green sector). For instance, MS and AD, which are neurological conditions more frequent in females than males, are localized in the top right of the picture. While females with MS demonstrated to have increased levels of both MMP-1 and MMP-10, females with AD have lower levels of MMP-3. On the contrary, males with PD, stroke, and epilepsy (right bottom side), have higher levels of MMP-1, MMP-9, and MMP-3, respectively. The situation is different for CVDs, where there is a more male-driven sex disparity, with aneurysm, MI, and CAD more frequent in this sex (left bottom). As such, several MMPs have been found to increase in males affected by these conditions. Finally, although HHD and RHD seem to be more frequent in females (top left side), results about MMP levels are still lacking (question mark symbol).

HHD: hypertensive heart disease; RHD: rheumatic heart disease; MS: multiple sclerosis; AD: Alzheimer’s disease; PD: Parkinson’s disease; CAD: coronary heart disease; MI: myocardial infarction.”

Abbreviations should be defined at first mention and used consistently throughout the manuscript (for example, line 158: “reactive oxygen species” should be “ROS”)

  1. We thank the Reviewer for pointing this out. We adjusted the abbreviation accordingly.

Reviewer 2 Report

In this review, Alessandro and colleagues summarized the findings of matrix metalloproteinases (MMPs) in cardiovascular and neurological diseases. Based on sex-related differences, the authors further estimated MMPs level changes in these diseases and brought up the idea of using MMPs as biomarkers in these diseases. The article provides an interesting scientific perspective on biomarker development for disease prognosis or progression, which should be of interest to clinical researchers in particular.

Some points to consider in subsequent versions:

1.       According to Figure 1, the overall neurological disease does not indicate a gender bias, while in Figure 4, nearly all the listed neurological diseases alone show gender bias.  To avoid reader confusions, the authors may need to explain this further.

2.       In Figure 2, to better demonstrate the MNPs family, I would suggest the author to name the classified MNPs or list the representative MNPs for each class.

3.       As a striking motor neuron disease, Amyotrophic lateral sclerosis (ALS) is more common in male than female (Manjaly, Z. R., et al, 2010. Amyotroph Lateral Scler 11(5): 439-442.), and it has been reported that MMP-9 plays a role in the pathogenesis of ALS (Kiaei M, et al, Exp Neurol. 2007;205:74–81.), (Kaplan A, et al. Neuron. 2014;81:333–348.), I would suggest the authors to incorporate these findings into the manuscript to enrich the summary of neurological disease.

4.       Some reference literatures are required for such statements in the manuscripts:

1)      Line 34-35, “sex differences are also able to influence both health and disease”

2)      Line 291-292, “serum MMP was positively correlated with inflammatory markers only in females”

Author Response

Reviewer 2

In this review, Alessandro and colleagues summarized the findings of matrix metalloproteinases (MMPs) in cardiovascular and neurological diseases. Based on sex-related differences, the authors further estimated MMPs level changes in these diseases and brought up the idea of using MMPs as biomarkers in these diseases. The article provides an interesting scientific perspective on biomarker development for disease prognosis or progression, which should be of interest to clinical researchers in particular.

Some points to consider in subsequent versions:

  1. According to Figure 1, the overall neurological disease does not indicate a gender bias, while in Figure 4, nearly all the listed neurological diseases alone show gender bias. To avoid reader confusions, the authors may need to explain this further.
  2. We thank the Reviewer for his/her suggestion. As he/she noted, gender bias in neurological diseases seems not present. This is largely due to the inclusion of the stroke category in neurological conditions. Usually, stroke is included within the CVDs given that the primary event is cardiovascular. However, the secondary and main affection is related to the brain and so it can also fall within the neurological conditions group. For this reason, we decided to include stroke in this group. As a reference, here is the figure including stroke in CVDs.

As you can see, although not so striking, the difference between sexes in neurological conditions is more visible.

As suggested, we further clarified this matter in the text (see lines 387-393): “Of note, the apparent counterintuitive lack of difference in neurological conditions showed in Figure 1 is mainly due to the inclusion of stroke-related diseases in this group. In fact, by leaving stroke within the CVDs the sex bias in neurological conditions appears to be more evident (DALYs*100,000 population [95% Uncertainty Level], males: 413 [251-646]; females: 565 [303-953]). For this reason, it is of paramount importance that the effect of sex is evaluated also separately in each condition, to avoid possible misinterpretations driven by grouping variables.”

  1. In Figure 2, to better demonstrate the MNPs family, I would suggest the author to name the classified MNPs or list the representative MNPs for each class.
  2. As suggested, we have now changed Figure 2 by including the class of each MMP with examples.

  1. As a striking motor neuron disease, Amyotrophic lateral sclerosis (ALS) is more common in males than females (Manjaly, Z. R., et al, 2010. Amyotroph Lateral Scler 11(5): 439-442.), and it has been reported that MMP-9 plays a role in the pathogenesis of ALS (Kiaei M, et al, Exp Neurol. 2007;205:74–81.), (Kaplan A, et al. Neuron. 2014;81:333–348.), I would suggest the authors to incorporate these findings into the manuscript to enrich the summary of neurological disease.

  1. We thank the Reviewer for his/her comment. Accordingly, a reference to the different sex distribution of ALS was placed in the text, and the involvement of MMP-9 in ALS pathogenesis was cited in the text and added in the new Table 1. The three citations suggested by the Reviewer were appropriately cited in the text and added to the bibliography. In particular:

1)  The sentence in lines 249-251 "MMP-9 was associated to demyelination in MS [73], to neuronal cell death in AD [74], and to neuroinflammation in PD and dolichoectasia [63,75,76]" has been changed to "MMP-9 was associated with demyelination in MS [82], neuronal cell death in AD [83], neuroinflammation in PD [72], dolichoectasia [84,85], and neuroinflammation and cell death in amyotrophic lateral sclerosis (ALS) [86,87].".

2) The sentence in lines 343-345: "The only exceptions appear to be Parkinson’s disease (PD), epilepsy and hemorrhagic stroke, where a male prevalence is documented [119–121]" has been changed to "The only exceptions appear to be Parkinson’s disease (PD), epilepsy, hemorrhagic stroke, and Amyotrophic Lateral Sclerosis (ALS), where a male prevalence is documented [137–140].

  1. Some reference literatures are required for such statements in the manuscripts:

1) Line 34-35, “sex differences are also able to influence both health and disease”

  1. Ok. Done

2)      Line 291-292, “serum MMP was positively correlated with inflammatory markers only in females”

R. Ok. Done

Round 2

Reviewer 1 Report

The manuscript improved after the revision. 

I only noticed that reference 98 and reference 113 are the same.